# Thermodynamic Analysis and Experimental Study on the Oxidation of PbX (X = S, Se) Nanostructured Layers

**DOI:** 10.3390/mi13081209

**Published:** 2022-07-29

**Authors:** Evgeniya Maraeva, Alexander Maximov, Nikita Permiakov, Vyacheslav Moshnikov

**Affiliations:** Department of Micro-and Nanoelectronics, Faculty of Electronics, Saint-Petersburg Electrotechnical University «LETI», 5, pr. Popova, 197022 Saint Petersburg, Russia; aimaximov@mail.ru (A.M.); nvpermiakov@etu.ru (N.P.); vamoshnikov@mail.ru (V.M.)

**Keywords:** phase equilibrium, oxidization, metal chalcogenides, Pb-S-O, Pb-Se-O

## Abstract

Heat treatment in an oxygen-containing medium is a necessary procedure in the technology of forming photodetectors and emitters based on lead chalcogenides. Lead chalcogenide layers (PbS, PbSe) were prepared via a chemical bath deposition method. Surface oxidation of lead chalcogenide layers was analyzed using X-ray diffraction and Raman spectroscopy methods, and thermodynamic analysis of the oxidation of PbSe and PbS layers was also performed. The calculated phase diagrams from 20 °C to 500 °C showed good agreement with the experimental results. According to the thermodynamic analysis, the oxidation products depend on the initial composition of the layers and temperature of the annealing. In some cases, the formation of a separate metallic phase Pb is possible along with the formation of lead oxide PbO and other oxides. The performed thermodynamic analysis makes it possible to substantiate the two-stage annealing temperature regimes which ensure an increase in the speed of photodetectors.

## 1. Introduction

Lead chalcogenides occupy an important place among semiconductor materials with desired functional properties. Possessing a unique set of electrical properties, they are materials for the manufacture of infrared detectors, chemical sensors, LEDs, solar cells, thermoelectrics, and supercapacitors [1,2,3,4,5,6,7]. It is known that the use of layers based on lead chalcogenides in infrared optoelectronics requires their sensitization by oxidation [8,9,10,11]. At the same time, the properties of the materials depend on the deviation from stoichiometry, and the defects in the materials can be redistributed during technological operations [12]. In this regard, there has been much attention paid to studying the influence of oxygen on the properties of structures based on lead chalcogenides [13,14,15,16].

The operational efficiency of the structures based on lead chalcogenides with highly photosensitive properties is obtainable by the creation of excessive charge carriers in a thin surface layer under the influence of light radiation and using the grain shell layer with a low concentration value of p-type charge carriers, close to the value corresponding to the stoichiometric composition, as a channel of conductivity.

In this case, a layer of a grain shell with a low concentration of p-type charge carriers close to the concentration corresponding to the stoichiometric composition is used as the conduction channel. Then, during the excitation, the concentration of charge carriers is increased repeatedly due to their injection from the grain core with n-type conductivity that causes high photosensitivity. Control of the achievement of such goals can be determined by the composition of the resulting oxides. There are experimental data, for example, in the case of the oxidation of lead selenide, that show that the layers containing dielectric phases such as lead selenites and oxyselenites [9,11] have a high photosensitivity.

At the same time, an urgent task is to increase the speed of the photodetectors. From the point of view of the formation of the oxides on semiconductor materials, we can use ideas about “fast” and “slow” recombination centers. The “slow states” occur inside the dielectric layer, and the “fast states” occur at the “dielectric–semiconductor” boundary.

As a way to increase the speed of the photostructures, it is possible to use the idea of creating “fast states” at the grain boundary of a narrow-gap semiconductor. The “fast states” can be metallic inclusions that provide efficient recombination of charge carriers or their removal from the phase interface. The lag of photodetectors depends on the presence or absence of “fast states” at the grain boundary of a narrow-gap semiconductor and on the thickness of the oxide phase layer. The creation of such states near the phase boundary is a technologically important operation for increasing photodetector speed. Currently, there is an active search for ways to increase the speed of photodetectors based on lead chalcogenides, particularly by combining layers of chalcogenides with carbon-containing particles [17]. We assume that the formation of pure metal (lead) nanoparticles can be a similar solution to this problem.

It follows from the experimental data that some authors have observed the appearance of metallic inclusions during annealing the layers of an oxygen-containing medium at high temperatures [18]. These results were generally interpreted as erroneous.

The aim of this work was to perform a thermodynamic analysis for the processes occurring in the Pb-X-O systems (X = S, Se) to estimate the probability of the formation of the metallic phase by the interaction of PbX with oxygen. Possible reactions in Pb-S-O and Pb-Se-O systems were considered, and partial pressure diagrams and Gibbs triangles were constructed at different annealing temperatures. In the experimental part, a chemical bath deposition technique was used for obtaining the layers, while XRD and Raman spectroscopy were applied for the sample characterization.

## 2. Materials and Methods

The thermodynamic analysis of the oxidation processes was carried out by the method of partial pressure diagrams [19,20] and the triangulation method [21,22,23]. The method of partial pressures allows us to establish manageable thermodynamic conditions for obtaining the necessary properties of the surface layers. The partial pressure diagrams were plotted in lgP_X_o_2_ − lgPo_2_ (X = S, Se) coordinates in the temperature range from 20 °C to 500 °C.

To obtain the layers of lead sulfide and lead selenide, we used the technology of chemical bath deposition developed at the Ural Federal University named after the First President of Russia B.N. Yeltsin [24]. This method is simple to implement and offers wide technologically important possibilities, which make it possible, owing to the colloidal chemical stage basic to it, to produce nanostructured semiconductor layers with controlled particle size. The layers were deposited on glass substrates in thermally resistant glass reactors. The substrate was fixed in a special 3D printed holder, at a certain angle with the working surface, and immersed in the prepared solution. Then, the reactor was placed in a thermostat, which was heated to the temperature required for the synthesis. The obtained layers were heat treated in an oxygen-containing medium by the method similar to that described in [25]. We prepared series of the samples of sulfide and lead selenide using the method of chemical bath deposition to compare the experimental data with the results of thermodynamic analysis. To obtain the layers based on lead sulfide, we used the following precursors: lead acetate Pb(CH_3_COO)_2_∙3H_2_O (GOST 1027-67); sodium citrate Na_3_C_6_H_5_O_7_∙2H_2_O (GOST 22280-76); ammonia water NH_3_∙H_2_O (GOST 24147-80); thiourea CS(NH_2_)_2_ (Sigma Aldrich, St. Louis, MO, USA). The initial solutions of reagents were prepared using distilled water. The layers were heat treated at the temperatures of 200 °C, 300 °C, 400 °C, 500 °C. To obtain the layers based on lead selenide, we used the following precursors: lead acetate Pb(CH_3_COO)_2_∙3H_2_O (GOST 1027-67); sodium citrate Na_3_C_6_H_5_O_7_∙2H_2_O (GOST 22280-76); ammonia water NH_3_∙H_2_O (GOST 24147-80); selenurea CH_4_N_2_Se (Sigma Aldrich). The thickness of the final layers after deposition was measured using an interference microscope (Linnik microinterferometer MII-4) and it amounted to approximately 100 nm.

To study the phase composition of the layers, we used X-ray diffraction analyses (Bruker-AXS and DRN Farad), as well as the Raman spectroscopy (RS) method. Raman spectra were recorded on a LabRamHR800 spectrometer combined with a confocal microscope (Jobin-Yvon Horiba). The second harmonic of the Nd:YAG-laser (wavelength 532 nm) served as the excitation source.

## 3. Results and Discussion

### 3.1. Reactions

To construct the partial pressure diagrams, a set of chemical reactions proceeding in the ternary Pb-S-O, Pb-Se-O systems was used. To find the numerical values of the equilibrium constant for reactions at a given temperature, the enthalpy, entropy, and Gibbs free energy were calculated for each reaction.

The main reaction equations which will probably take place in the Pb-S-O system are shown as the following:Pb (s) + 0.5 O_2_ (g) = PbO (s)(1)
PbS (s) + 1.5 O_2_ (g) = PbO (s) + SO_2_ (g)(2)
PbS (s) + O_2_ (g) = Pb (s) + SO_2_ (g)(3)
PbO (s) + SO_2_ (g) + 0.5 O_2_ (g) = PbSO_4_ (s)(4)
PbO (s) + (1/6)O_2_ (g) = (1/3) Pb_3_O_4_(s)(5)
(1/3) Pb_3_O_4_ (s) + SO_2_ (g) + (1/3) O_2_ (g) = PbSO_4_(s)(6)
PbO_2_ (s) + SO_2_ (g) = PbSO_4_ (s)(7)
PbS (s) + 2O_2_ (g) = PbSO_4_ (s)(8)
S (s) + O_2_ (g) = SO_2_ (g)(9)
S (s) + O_2_ (g) = SO_2_ (s)(10)
SO_2_ (g) = SO_2_ (s)(11)
SO_2_ (s) + 0.5O_2_ (g) = SO_3_ (s)(12)
SO_3_ (s) = SO_2_ (g) + 0.5 O_2_ (g)(13)

The main reaction equations which will probably take place in the Pb-Se-O system are shown as the following:PbO (l) = Pb (l) + 0.5 O_2_ (g),(14)
PbSe (s) + O_2_ (g) = Pb (l) + SeO_2_ (g)(15)
PbSe (s) + 1.5O_2_ (g) = PbO (l) + SeO_2_ (g)(16)
PbSe (s) + 1.5O_2_ (g) = PbSeO_3_ (s)(17)
3PbSeO_3_ (s) = 2PbO∙PbSeO_3_ (s) + 2SeO_2_ (g)(18)
5(2PbO∙PbSeO_3_) (s) = 3(4PbO∙PbSeO_3_) (s) + 2SeO_2_ (g)(19)
4PbO∙PbSeO_3_ (s) = 5PbO (l) + SeO_2_ (g)(20)
PbSeO_4_ (s) = PbSeO_3_ (s) + 0.5O_2_ (g)(21)
3PbSe (s) + 4.5O_2_ (g) = 2PbO∙PbSeO_3_ (s) + 2SeO_2_ (g)(22)
5PbSe (s) + 7.5O_2_ (g) = 4PbO∙PbSeO_3_ (s) + 4SeO_2_ (g)(23)
2PbSeO_4_ (s) = PbO∙PbSeO_4_ (s) + SeO_2_ (g) + 0.5O_2_ (g)(24)
5(PbO∙PbSeO_4_) (s) = 2(4PbO∙PbSeO_4_) (s) + 3SeO_2_ (g) + 1.5O_2_ (g)(25)
4PbO∙PbSeO_4_ (s) = 5PbO (l) + SeO_2_ (g) + 0.5O_2_ (g)(26)
3(PbO∙PbSeO_4_) (s) = 2(2PbO∙PbSeO_3_) (s) + SeO_2_ (g) + 1.5O_2_ (g)(27)
3(4PbO∙PbSeO_4_) (s) + 2SeO_2_ (g) = 5(2PbO∙PbSeO_3_) (s) + 1.5O_2_ (g)(28)
4PbO∙PbSeO_4_ (s) = 4PbO∙PbSeO_3_ (s) + 0.5O_2_ (g)(29)
2PbSeO_3_ (s) + 0.5O_2_ (g) = PbO∙PbSeO_4_ (s) + SeO_2_ (g)(30)

We used a set of presented reactions (1)–(30) to construct partial pressure diagrams and to identify the stability regions of a particular condensed phase depending on the composition of the gas phase.

### 3.2. The Thermodynamic Analyses Data

As was shown in some of our previous papers, e.g., [26], the results of thermodynamic analysis of the Pb-S-O system confirm the possibility of the formation of pure lead at high temperatures of annealing of lead sulfide. Figure 1, Figure 2 and Figure 3 present partial pressure diagrams and diagrams of coexisting phases (Gibbs triangles) in the Pb-S-O system.

The three-phase equilibrium points in the figures (a) correspond to the region of coexistence of three phases on the Gibbs triangles (b). As can be seen from Figure 1, Figure 2 and Figure 3, at a temperature above 500 K, the quasi-binary section “PbS–PbO” that existed in Figure 1b disappears and the quasi-binary section “Pb–PbO∙PbSO_4_” becomes possible (Figure 3b).

Thus, a certain increase in the annealing temperature of the PbS layers, even for the layers with an insignificant deviation from stoichiometry, will lead to the release of free lead at the interface of the oxide phase and lead sulfide grain, along with the formation of the PbO·PbSO_4_ phase. The changes in the quasi-binary sections occur through the passage of a four-phase equilibrium situation (Figure 2b). As the thermodynamic analysis showed, this situation occurs at the temperature of 217.5 °C.

As can be seen from Figure 1a,b, at low temperatures, lead sulfide is in highly non-equilibrium conditions in relation to the partial pressure of oxygen (0.21 atm). The composition of oxide phases on the surface of lead sulfide significantly depends on the deviation from stoichiometry. At significant concentrations of n-type charge carriers (excess of lead), the oxide layer corresponds to the PbO phase. When the deviation from stoichiometry is reduced, the oxide composition becomes PbO∙PbSO_4_ and PbSO_4_. Figure 2 corresponds to the four-phase equilibrium temperature. With a further increase in temperature (Figure 3a,b), the PbO∙PbSO_4_ phase forms on the n-type PbS surface. At the same time, taking into account the kinetics of formation of the oxide layer at lower temperatures, the following reaction can proceed at the interface of the PbS—PbO phases:PbS (s) + 5PbO (s) = 4Pb (s) + PbO∙PbSO_4_ (s).(31)

It should be noted that with an increase in the thickness of the oxide created at the initial stage, the kinetics of oxidation of lead precipitates at the interface will be slowed down and the presence of inclusions of pure lead may be observed for a long time. This explains the paradoxical fact of the detection of the phase of pure lead by X-ray photoelectron spectroscopy during the oxidation of lead sulfide in [18] at high annealing temperatures.

In the same way, we analyzed the partial pressure diagrams in the Pb-Se-O system. We found that the type of diagrams in this system in the temperature range from room temperature to the melting point of lead selenide does not change. Figure 4a,b, for example, shows a partial pressure diagram calculated in the Pb-Se-O system at the temperature of 400 °C and a diagram of coexisting phases characterizing the Pb-Se-O system at this temperature.

Unlike the Pb-S-O system, in the Pb-Se-O system the formation of pure lead at the interface of the oxide and chalcogenide phase of lead is not thermodynamically advantageous. From this point of view, lead selenide can be a more preferable material for creating photoemitters, since in this system there are no nonradiative recombination centers due to the presence of pure lead.

### 3.3. Investigation of Deposited PbX (X = S, Se) Layers

Figure 5a,b show the fragments of the X-ray diffraction pattern obtained in the study of lead sulfide layers formed by chemical bath deposition on glass substrates before and after heat treatment.

As can be seen from Figure 5, in the absence of annealing of the initial lead sulfide layers, the X-ray pattern shows the lines corresponding to the PbS, nPbO·PbSO_4_ (n = 2; 4) phases. Temperature action leads to a greater degree of crystallization of the PbS grains; at high-temperature annealing, the peaks of reflections are narrow, which means an increase in the size of crystallites. The oxide phase PbO∙PbSO_4_ forms already at room temperature. At the temperatures of heat treatment above 300 °C, separate reflections corresponding to the phase of crystalline lead begin to appear on the X-ray diffraction pattern. This fact corresponds to the data on thermodynamic analysis.

In the study of the surface composition of the layers based on lead selenide, the method of Raman spectroscopy was additionally applied. When studying the layers by the Raman method, the laser beam was focused on a spot with a diameter of ~ 1–2 μm on the sample surface. The typical power density did not exceed 5 kW/cm2 in order to avoid the influence of laser irradiation on the structure of the objects under study. For example, Figure 6 shows the Raman spectra of a layer of lead selenide before heat treatment in comparison to a layer annealed at the temperature of 400 °C.

The Raman spectrum of the lead selenide layer before annealing is presented by broad bands with maxima near 100 cm–1, 125 cm–1, and 230 cm–1 (Figure 6). According to the data of [27], these bands correspond to the PbSe phase.

The Raman spectrum of the lead selenide layer annealed at T = 400 °C is also presented by broad bands with maxima near 100 cm^−1^, 125 cm^−1^, and 230 cm^−1^. Additionally, one more clear peak can be seen from the Raman spectrum (790 cm^−1^), presumably corresponding to the PbSeO_4_ phase [28]. As can be seen in the Raman spectrum of the initial layer which was not heat treated (Figure 6a), no additional peak was observed before this treatment was performed.

Figure 7 shows a fragment of the schematic X-ray diffraction pattern of the PbSe sample that has been heat treated at the temperature of 500 °C.

It can be seen from Figure 7 that there are peaks in the diagram corresponding to the PbSe phase; this suggests that the method of chemical bath deposition makes it possible to obtain the layers with a well-formed crystalline structure. Additionally, the PbSeO_3_ phase is present. This oxide phase corresponds to the stoichiometric composition of the layer (Figure 4b). In addition, the presence of a complex oxide phase PbO∙PbSeO_4_ was recorded. The results obtained are also consistent with the data of preliminary thermodynamic analysis.

## 4. Conclusions

Thermodynamic analysis of lead sulfide oxidation showed that the absence of heat treatment, as well as heat treatment at temperatures below 500 K, in the case of layers with a small deviation from stoichiometry, leads to the formation of lead oxysulfate PbO∙PbSO_4_ and lead oxide PbO. During heat treatment above 500 K, in the event of a significant deviation from stoichiometry towards an excess of lead, the formation of a separate metallic phase Pb is also possible along with the formation of lead oxide PbO and other oxides.

Thermodynamic analysis of oxidation processes in the Pb-Se-O system showed that in the temperature range from room temperature to the melting point of lead selenide, the formation of pure lead at the interface of the oxide and chalcogenide phase of lead is not thermodynamically advantageous.

Surface oxidation of the lead chalcogenide layers was investigated by X-ray diffraction and Raman spectroscopy methods. The X-ray spectra of lead sulfide after heat treatment at the temperature of 300 °C recorded the presence of a phase of pure lead, along with oxide phases. X-ray diffraction analysis of lead selenide layers after heat treatment in an oxygen-containing medium showed the presence of oxide phases only. The additional studies via the Raman spectroscopy method showed the presence of the PbSeO_4_ phase in this case.

It should also be noted that in nanocomposites, due to the size effects, the temperatures corresponding to a change in equilibria between the phases may change. However, in general this thermodynamic approach allows us to understand the physicochemical nature of the observed “anomalous” effects of the appearance of unoxidized lead microinclusions at high temperatures, and also to substantiate the two-stage temperature annealing regimes which ensure an increase in the speed of photodetectors due to “fast” recombination processes through the levels of the metal at the oxide–semiconductor interface. The choice of annealing temperature regimes during the formation of the photosensitive layer for photodetectors at the first (low-temperature) stage can create a protective oxide layer in the absence of metal inclusions. At a higher annealing temperature (second stage), the initial oxide, when interacting with a semiconductor, can lead to the appearance of metallic inclusions and oxides of a different composition. In combination with a photoemitter, such photodetectors are successfully used in optocouplers for gas detection.

## Figures and Tables

**Figure 1 micromachines-13-01209-f001:**
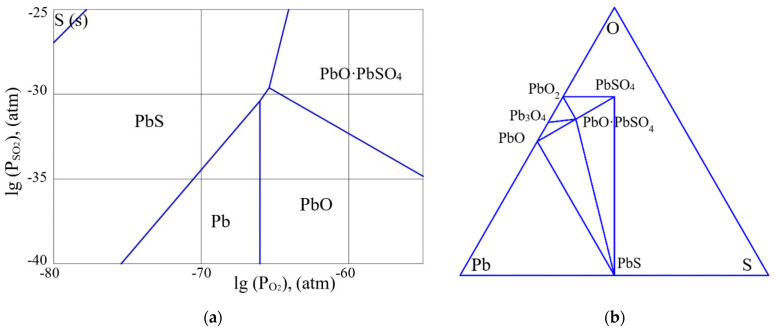
(**a**) The partial pressure diagram in the Pb-S-O system at the temperature of 20 °C. (**b**) The diagram of coexisting phases in the Pb-S-O system at the temperature of 20 °C.

**Figure 2 micromachines-13-01209-f002:**
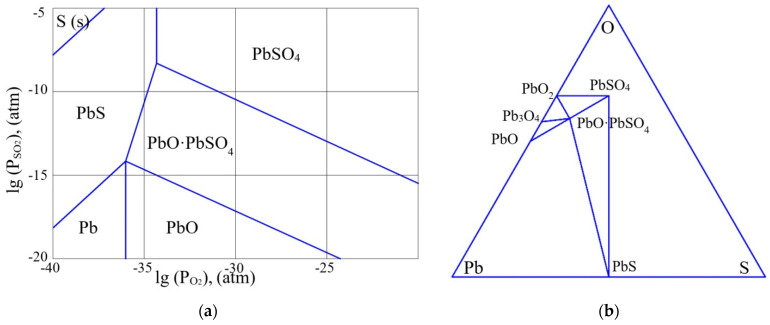
(**a**) The partial pressure diagram in the Pb-S-O system at the temperature of 20 °C. (**b**) The diagram of coexisting phases in the Pb-S-O system at the temperature of 217.5 °C.

**Figure 3 micromachines-13-01209-f003:**
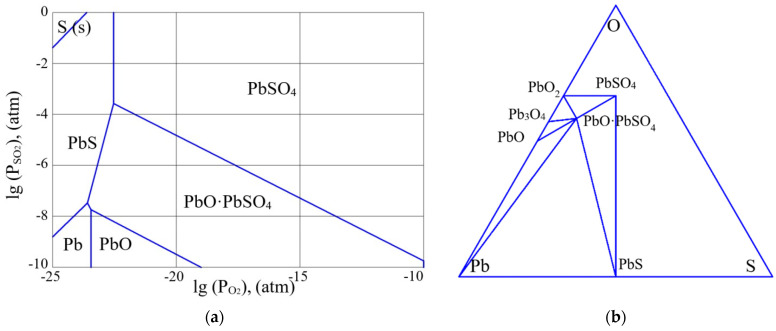
(**a**) The partial pressure diagram in the Pb-S-O system at the temperature of 20 °C. (**b**) The diagram of coexisting phases in the Pb-S-O system at the temperature of 400 °C.

**Figure 4 micromachines-13-01209-f004:**
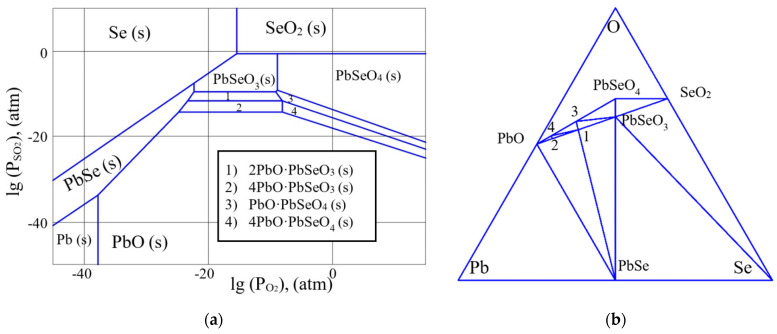
(**a**) The partial pressure diagram in the Pb-Se-O system at the temperature of 400 °C. (**b**) The diagram of coexisting phases in the Pb-Se-O system at the temperature of 400 °C.

**Figure 5 micromachines-13-01209-f005:**
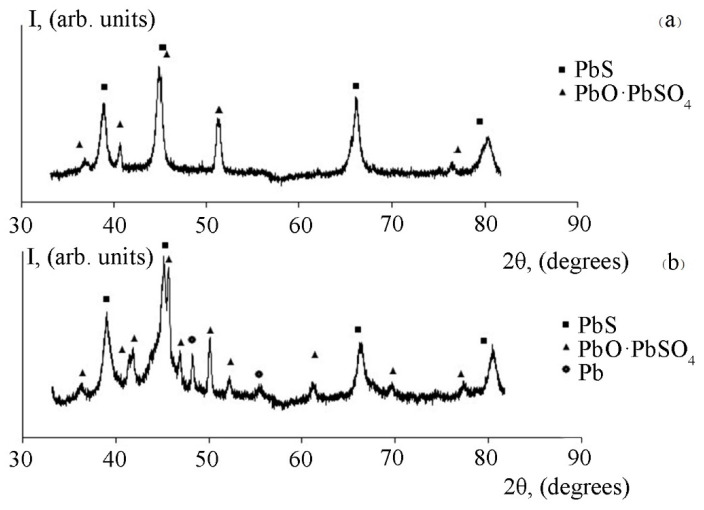
(**a**) A fragment of the X-ray diffraction pattern of the lead sulfide layer before annealing, reflexes for PbS (from left to right)—(111), (200), (220), (311) and reflexes for PbO∙PbSO_4_ (from left to right)—(201), (310), ($11\bar{2}$), ($22\bar{1}$), (222). (**b**) A fragment of the X-ray diffraction pattern of the lead sulfide layer annealed at the temperature of 300 °C, reflexes for PbS (from left to right)—(111), (200), (220), (311) and reflexes for PbO∙PbSO_4_ (from left to right)—(201), (310), (002), ($11\bar{2}$), ($31\bar{2}$), (021), ($22\bar{1}$), ($31\bar{3}$), ($60\bar{3}$), (222), reflexes for Pb (from left to right)—(111), (200).

**Figure 6 micromachines-13-01209-f006:**
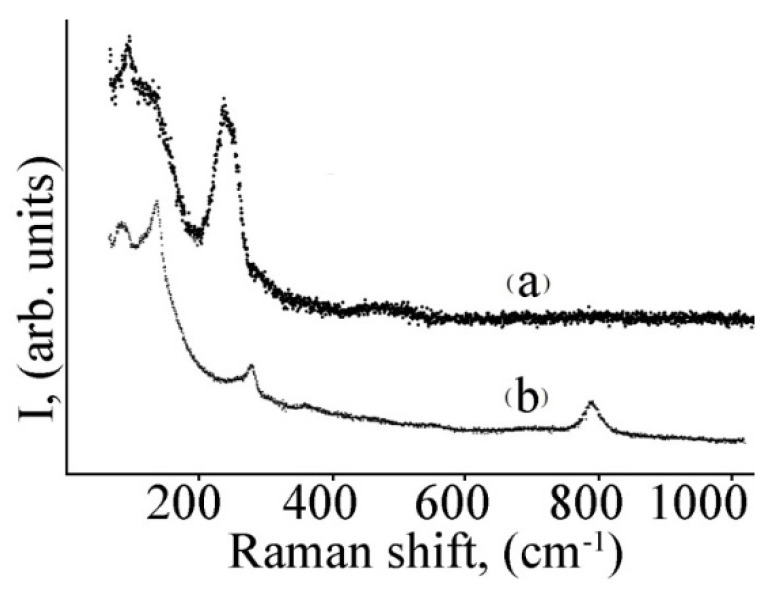
Raman spectra of lead selenide layer before heat treatment (**a**) and annealed at the temperature of 400 °C (**b**).

**Figure 7 micromachines-13-01209-f007:**
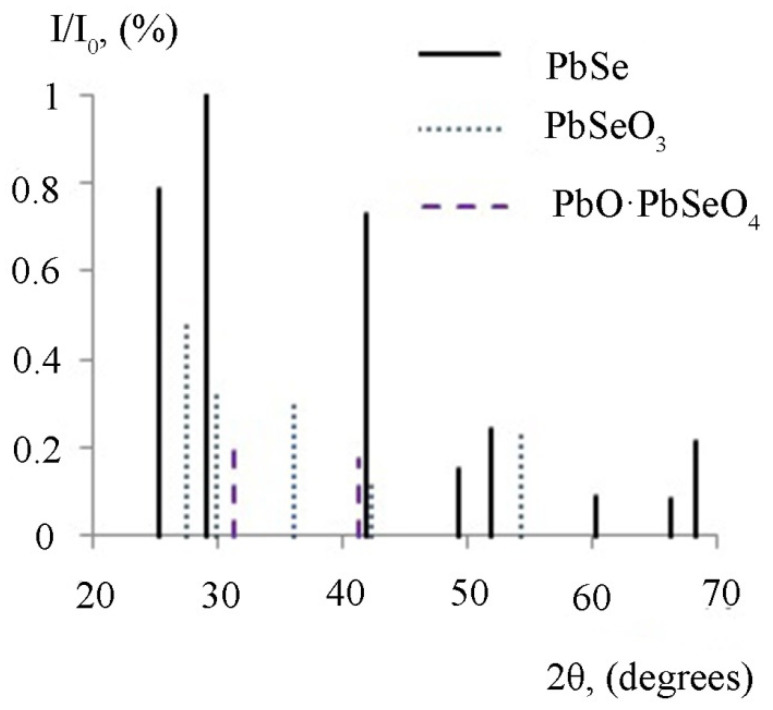
A fragment of the schematic X-ray diffraction pattern of the lead selenide layer annealed at the temperature of 500 °C.

## Data Availability

Not applicable.

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
