# Peer review of "Thermodynamic Analysis and Experimental Study on the Oxidation of PbX (X = S, Se) Nanostructured Layers"

_micromachines, 2022, doi:10.3390/mi13081209_

Round 1

Reviewer 1 Report

The manuscript appears to be mathematically and experimentally correct, and the utilized methods are interesting. I recommend the paper for publication. However, there are some concerns, comments, and suggestion should be addressed before publication:

(1) The abstract describes well the research program but does not mention the key findings and conclusions of the study. It is important to include some of these in the abstract.

(2) In the introduction part, the authors present several references where the work developed is described, but in very few situations, the main conclusions are summarized. Under these conditions, this section does not help the authors in the discussion of their results, nor does it clarify the readers on the studied subject. On the other hand, this section is very vague and the novelty of this study compared to those in the literature is not evident. Therefore, the novelty is neither clear nor evident, and I suggest a significant improvement.

(3) The references of the articles are old, most of them are articles from more than ten years ago, and there are few references from the past five years. The authors are advised to cite recent literature, especially the ones published in this journal.

(4) The whole article should be rechecked because of grammatical problems.

(5) The results and figures are appropriate. However, the authors should add more physical explanations for the observed results.

Reviewer 2 Report

Evgeniya Maraeva et al. prepared 10 layers (PbS, PbSe) by chemical bath deposition. Then, they demonstrated surface oxidation evolution using X-ray diffraction and Raman spectroscopy. In addition, relevant thermodynamic analysis was conducted, respectively. The experimental results have a good consistance with calculated phase diagrams.  After considering the following points, I think it could be accepted.

1. The authors should give a graphic abstract ( such as chemical structures of the layers) to highlight their findings.

2. It is important to measureor control the thickness of each layer or final whole layer, because size may have a key effect on the oxidation behavior and thermodynamic analysis.

3. If possible, the auhtors provide more potent applications of this work (such as semiconducting devices) in the part of Conclusions.

Reviewer 3 Report

The paper Thermodynamic Analysis and Experimental Study on The Oxidation of PbX (X=S, Se) Nanostructured Layers is devoted to preparation and investigations of the lead chalcogenide layers (PbS, PbSe). Chemical bath deposition technique was used for the layers obtaining. XRD and Raman spectroscopy were applied for the samples characterization. The topic of this paper is critically actual especially in area of materials for the electronics application. The work is of scientific interest to the audience of the Micromachines. The data are reliable and do not cause much doubt. Nevertheless, there are several points before the paper can be published. I hope that authors after major revisions can improve the paper and can publish it in Micromachines.

  1. In the abstract, authors should add one sentence briefly about the significance of the work and broadly the area in which it is making an impact.
  2. The information about practical applications of the presented materials is absent.
  3. Why did you choose the chemical bath deposition technique for the samples preparation? Does it have advantages compare to the traditionally used techniques?
  4. The authors may try to improve the flow of the paper, especially in the introduction section. It can help the readers to understand the aim of each experiment and connections between paragraphs.
  5. The Introduction part must be improved with new relevant literature. I suggest using the following literature [please see and discuss:

https://doi.org/10.1007/s11664-021-08828-5;

https://doi.org/10.1016/j.jallcom.2021.161451;

https://doi.org/10.1088/1757-899X/848/1/012089;

https://doi.org/10.1002/pssb.201451013].

  1. “The thermodynamic analysis of oxidation processes was carried out by the method of partial pressure diagrams and the triangulation method.” – insert the references about these methods.
  2. Figure 5 – identify all XRD patterns (indexes).
  3. Conclusion must be re-written more widely.
  4. English must be improved, because now there are some typos and grammatical errors.

Round 2

Reviewer 1 Report

I recommend the paper for publication. 

Reviewer 3 Report

Now the paper can be accepted for publication